# Wearable Optical Fiber Sensors in Medical Monitoring Applications: A Review

**DOI:** 10.3390/s23156671

**Published:** 2023-07-25

**Authors:** Xuhui Zhang, Chunyang Wang, Tong Zheng, Haibin Wu, Qing Wu, Yunzheng Wang

**Affiliations:** 1Heilongjiang Province Key Laboratory of Laser Spectroscopy Technology and Application, Harbin University of Science and Technology, Harbin 150080, China; zxh_99ok@163.com (X.Z.); wangchunyang.work@outlook.com (C.W.); woo@hrbust.edu.cn (H.W.); 2School of Artificial Intelligence, Beijing Technology and Business University, Beijing 100048, China; 20211206@btbu.edu.cn; 3Center for Optics Research and Engineering, Shandong University, Qingdao 266237, China

**Keywords:** optical fiber sensors, wearable sensors, fiber Bragg grating, healthcare

## Abstract

Wearable optical fiber sensors have great potential for development in medical monitoring. With the increasing demand for compactness, comfort, accuracy, and other features in new medical monitoring devices, the development of wearable optical fiber sensors is increasingly meeting these requirements. This paper reviews the latest evolution of wearable optical fiber sensors in the medical field. Three types of wearable optical fiber sensors are analyzed: wearable optical fiber sensors based on Fiber Bragg grating, wearable optical fiber sensors based on light intensity changes, and wearable optical fiber sensors based on Fabry–Perot interferometry. The innovation of wearable optical fiber sensors in respiration and joint monitoring is introduced in detail, and the main principles of three kinds of wearable optical fiber sensors are summarized. In addition, we discuss their advantages, limitations, directions to improve accuracy and the challenges they face. We also look forward to future development prospects, such as the combination of wireless networks which will change how medical services are provided. Wearable optical fiber sensors offer a viable technology for prospective continuous medical surveillance and will change future medical benefits.

## 1. Introduction

Optical fiber sensors have been applied in many fields, such as engineering construction monitoring, defense, medicine, industry, and many other fields [1,2]. In agriculture, it can efficiently and quickly detect pesticide residues in farmland on site [3]. Optical fiber sensor has the character electromagnetic interference characteristicsility, small size, flexibility, multiplexing ability, and high accuracy [1]. In addition, because no current is required at the point of measurement, the technology provides the security needed to operate in wet or humid environments, such as sweating or water-based physical therapy procedures [2,4,5,6,7]. These features provide reliable technical support for continuous monitoring devices in physical applications.

Wearable optical fiber sensors, as a specialized category of optical fiber sensors, have garnered increasing attention in medical applications. This is primarily due to their exceptional flexibility, ease of wear, and high precision. These unique features make them highly suitable for continuously monitoring physiological parameters in medical settings [2,4,6,8,9,10,11]. With the continuous improvement of living standards and medical conditions, the life expectancy of human beings continues to rise. An aging population will lead to an increase in geriatric diseases, such as cardiovascular disease [12,13,14,15], stroke [16], Parkinson’s disease [17,18,19], and so on. Young individuals often engage in prolonged sedentary behavior, including extensive overtime work, late-night study sessions, and extended sitting. Unfortunately, adopting improper sitting postures can significantly increase the risk of developing various health issues, such as shoulder ailments [20], neck problems [21], and lower back conditions [22,23,24]. Rehabilitation for some diseases requires continuous monitoring of patient physiological parameters. These conditions have given impetus to the development of medical devices for continuous monitoring. 

Rehabilitation for certain conditions necessitates continuous monitoring of patients’ physiological parameters, driving the demand for developing medical devices tailored to continuous monitoring. This focus on continuous monitoring devices has imposed higher requirements on sensors, emphasizing the need for flexibility, compactness, and reliability to align with constant monitoring equipment. This focus on serial monitoring devices has set higher requirements on sensors, emphasizing the need for flexibility, compactness, and reliability to align with continuous monitoring equipment [20,21,23,25]. Portable electrochemical sensors can also be used in biological monitoring [26]. However, traditional sensors are generally electronic or electromechanical, which have low compactness, and need frequent calibration [1,24,27]. In addition, electronic devices are susceptible to electromagnetic interference and have electrical safety problems. By integrating optical fiber sensors into wearable devices, continuous monitoring devices can be designed, resolving the issues above associated with traditional sensors. 

Wearable devices based on fiber optic sensors have proven to be effective and comfortable solutions for monitoring vital signs (ankle [23,28,29,30,31,32,33], blood [22,34,35], spine [36], heart rate [22,37,38,39,40], breathing [10,22,41,42,43,44], shoulder and neck [45], joint angle [21,29,30,37,42,46], etc.). The characteristics of optical fibers allow them to be embedded in textiles such as T-shirts and elastic belts for sensing applications, enabling respiratory monitoring [47]. Wearable optical fiber sensors based on fiber Bragg gratings (FBG) can achieve an average error of 0.33% in respiratory monitoring, but they tend to be more expensive. On the other hand, wearable optical fiber sensors based on changes in light intensity have a more straightforward structure and lower cost but slightly lower accuracy in respiratory monitoring. They can also be integrated into elastic gloves to monitor finger joint angles for human–machine interaction [48]. In the field of motion analysis, wearable optical fiber sensors have been applied to monitor joint arches and gait [1]. Wearable optical fiber sensors based on Fabry–Perot interferometry (FPI) have achieved an accuracy of 0.0296 ± 0.001 mw/° in gait analysis. These wearable devices do not interfere with patients’ normal daily life or limit their range of movement. Accuracy of such physical enablers must always be guaranteed, as they are responsible for the overall performance of the wearable device and the data provided by medical staff for diagnosis. Through the wearable fiber optic sensor, the function of fiber optic technology can reach any part of the body and continuously monitor the physiological parameters of the human body [2,22,49,50,51,52,53,54,55]. The use of wearable devices, combined with recent advances in wireless technology, is expected to provide an appropriate solution for remote physical rehabilitation, allowing patients to perform the controlled therapeutic exercise from the comfort of their homes and possibly under constant remote supervision by a doctor or medical staff [23]. In addition, such solutions can be used continuously in daily life to monitor the progress of rehabilitation in daily activities and to continuously monitor disease conditions for prevention. The solution is expected to significantly reduce the influx of patients into hospitals and medical centers [1,6,10,22,56,57,58,59,60]. Therefore, reducing the costs associated with such physical therapy. In addition, storing medical data for further analysis helps to understand a patient’s case and health trends within a particular community [57,61,62,63]. This will significantly improve the level of social public medical care.

This article explores the various applications of wearable optical fiber sensors in medical monitoring. It provides a detailed introduction to the applications of wearable optical fiber sensors in respiratory monitor and joint monitoring, comparing and summarizing the working principles of three f wearable optical fiber sensors. Additionally, it discusses methods to improve accuracy in respiratory monitoring and joint monitoring, as well as the performance optimization of encapsulation materials. The paper begins by discussing the working principle of wearable optical fiber sensors based on fiber Bragg gratings and their wide-ranging applications in the medical field. It then provides an overview of fiber sensors utilizing changes in light intensity and examines their applications in healthcare. Finally, it discusses the working principle of fiber sensors based on Fabry–Perot interferometry and its application in the medical field. Wearable optical fiber sensors can revolutionize the future of medical consultations by enabling continuous monitoring of patients’ physiological parameters through wearable medical devices, eliminating the need for frequent hospital visits for check-ups. These devices are equipped with easily wearable optical fiber sensors, allowing patients to conveniently obtain vital physiological information without leaving their homes. By expediting patients’ recovery and fostering significant improvements in the healthcare environment, these sensors play a crucial role.

## 2. Wearable Fiber Bragg Grating Sensors

FBG [64] sensor is a thin fiber bundle composed of a fiber core, cladding layer, and buffer layer, and the fiber core is uniformly engraved with a lattice pattern. The carved lattice pattern reflects only specific wavelengths when the core emits broadband light. This reflection wavelength is called the Bragg wavelength and represents the relationship between the effective refractive index of the fiber core and the distance from one lattice to the next [36,65,66].
(1)λ=2nΛ
where λ is the Bragg wavelength, n is the effective refractive index, and Λ is the grating period. When the measured physical quantity (such as temperature, stress, etc.) Used for fiber grating changes, it will cause the corresponding change of n and Λ, resulting in λ drift and, conversely, by detecting the drift of λ [67]. Information about the measured physical quantity can also be obtained. Research on Bragg fiber grating sensors mainly focuses on quasi-distributed measurement of temperature and stress. Λ shift caused by temperature and stress changes can be expressed as:(2)Δλ=αΔT
(3)Δλ=εS
where Δλ is the wavelength change, α is the temperature coefficient of the fiber, ΔT is the temperature change, ε is the strain of the fiber, and S is the strain sensitivity of the grating. 

FBG is a hot optical fiber sensor inscribed virtually on any fiber. FBG arrays can be easily fabricated, with spacing ranging from several kilometers down to the millimeter [22,29,37,48,67,68,69,70,71]. FBG sensors are used in many industrial areas to monitor physical variables such as temperature, pressure, etc. In addition, FBG is mainly used to monitor physiological parameters and is more widely used in the medical field, such as blood pressure [34,72,73,74,75,76] and heartbeat [37,77,78,79,80,81], biomechanical research [9,53,82], temperature monitoring [9], and respiratory monitoring system [10,22,41,42]. FBG sensors have good measurement performance in tactile perception. In addition, their biocompatibility, non-toxicity, and chemical inertness, as well as their small size and flexibility, make them particularly suitable for invasive measurements in vivo. FBG’s immunity to electromagnetic interference and inherent magnetic resonance compatibility makes it the hottest application in wearable fiber sensors [37,83,84,85].

### 2.1. FBG Respiratory Monitoring

Respiration is an essential parameter of vital signs monitoring. In recent years, wearable devices for respiratory monitoring have become a general application due to the COVID-19 pandemic. Usually, the person’s breath is divided into the chest and abdominal; the chest and abdomen swell when inhaling, and the chest and abdomen sag when exhaling, forming a breathing movement. The principle of FBG respiratory monitoring equipment based on chest and abdomen movement is that the chest and abdomen displacement driven by respiratory activity leads to λ shift, which is transmitted to FBG. This characteristic is very suitable for FBG, which is usually packaged in a flexible material (Figure 1) because the chest and abdomen-based respiratory monitoring devices must fit the FBG well into the chest and abdomen [85,86,87,88,89]. Otherwise, it will cause errors. The packaging material is wrapped in the chest and abdomen position, and the chest and abdomen displacement in breathing movement will cause the deformation of the packaging material. The sensor obtains the maximum and minimum deformation at the end of inhalation and exhalation, respectively [90,91,92,93]. The separation and duration of the two deformations can be calculated to obtain the breathing frequency and cycle [94]. 

There are several FBG breathing monitors based on the chest and 141 abdominal movements and a new airflow-based FBG breathing monitor. M.C. et al. used two commercial FBGs in one fiber [63] (Figure 2a). The optical fiber is fixed to a commercial slim-fit t-shirt in two positions through a thick silicone rubber containing about 15 cm^2^ of trajectory (methyl) silane. To record the chest movement, the subjects placed 19 passive light reflective markers on the horizontal plane of the clavicular line, the posterior hand joint, and the nipple. System performances, in terms of respiratory period, duration of inspiratory and expiratory phases, as well as left and right upper thorax (UT) volumes, were assessed on four healthy volunteers, and the sensitivity is always higher than 0.66 and 0.35. The disadvantage of the design is that the glass fiber optics used are fragile. Massaroni’s team proposed an intelligent textile based on six FBGs [50] (Figure 2b). The optical fiber is fixed in the commercial slim l t-shirt at six resettlement sites with adhesive trimethoxysilane silicone rubber. Then, 42 emission light reflective markers with a diameter of 9 mm are placed in front of the chest wall of the subject’s skin. During quiet breathing and maximum inspiratory action, the FBG experienced about 0.2 nm and 0.4 nm, the maximum change of b value, the maximum difference between the two Br values of subjects is 0.29 beats per minute (BPM), which is about 1.59%. A new method is adopted to optimize the positioning of FBG on intelligent textiles to improve each FBG’s sensitivity in monitoring compartment volume changes. Optoelectronic plethysmography is used to collect the reference respiratory chamber volume, rather than spirometry as the reference instrument of lung volume. The Carlo team used viscous silicone rubber containing thimerosal (methyl silane) to Bond 12 FBG sensors (with a grating length of 10 mm) to a T-shirt [51] (Figure 2c). The 12 FBGs allowed local strain analysis of six functionally separable chambers of the chest wall (left and right sides of the lung chest, abdominal chest, and abdomen). A four-channel spectral interrogator collects the output at a sampling rate of 250 Hz. Using hypoallergenic tape, 89 infrared light reflection markers are placed on the intelligent textiles worn by volunteers, and the data collected by the textiles are materials with the data obtained by the motion analysis system (used as a reference instrument). The two systems showed good consistency in the respiratory cycle (bias of 0.01 s), respiratory frequency tendencies of 0.02 breaths·min^−1^), and tidal volume (bias of 0.09 L).

D. Lo Presti’s team first combined neck motion monitoring and respiratory monitoring to achieve the monitoring of two parameters with one device [45] (Figure 3a). It is composed of two flexible sensors based on FBG Technology. Each sensor encapsulates an FBG in a rectangular matrix made of silicone rubber. The use of polyacrylate bandages can better adhere to and conform to the skin. This bandage has viscosity, elasticity, and high permeability; breathing can be monitored through neck muscle activity and chest junction movement. During the tests, the participants performed five repetitions of flexion, extension, and axial rotation and ten repetitions of profound and rapid breathing. The output variations of the wearable device and the motion capture system collected during the flexion-extension (FE) and axial rotation (AR) repetitions are shown in Figure 3b. The system performs well in monitoring the aforementioned parameters compared to the motion capture system. All signals involved in peak detection for breath analysis are depicted in Figure 3c,d. The output variations of each volunteer, collected by the flow meter and FBG, are displayed during quiet and rapid breathing. The wearable system effectively captures the temporal trends of neck movements (flexion, extension, and axial rotation). It accurately estimates the average breath rate and breath rate values for both quiet breathing and rapid breathing. The percentage errors are 6.09% and 1.90%, respectively.

Aizhan misstated’s team designed a T-bar embedded with two belts [22] (Figure 4a). One strap is fixed on the chest, and the other is set on the abdomen. Each belt has five FBG arrays, a total of 10 measurement points. Two subjects are monitored for breathing in four positions: stay, sit, lie, and run. The accuracy of the results of the two volunteers is different but remains the same. Multipoint sensing improves the system’s accuracy, and the four postures monitored by the system align with daily habits. J. Di Tocco’s team innovatively encapsulated FBG in a dumbbell-shaped silicone matrix (Figure 1b) rather than directly integrating it into the elastic fabric. They selected silicone resin as the shell material to improve the extensibility. Considering the influence of temperature on FBG respiratory monitoring equipment for the first time, the innovation of polymer packaging material has dramatically improved the performance of wearable optical fiber sensors. Packaging FBG in a silicone matrix cannot only protect FBG from the impact of the external environment, but also the low thermal conductivity of silicone material will significantly improve the accuracy of FBG and reduce the impact of temperature on FBG. The design of the dumbbell shape will make the stress more concentrated, using polyimide as the fiber layer dramatically improves the durability and flexibility of the fiber. The experimental data of polymer-coated and encapsulated FBG in tensile and temperature tests are highly consistent with the linear fitting. The sensing element is fixed on the elastic belt through the hook and loop anchorage system [47] (Figure 4b). The subjects wore the device on their chests and abdomens and underwent testing through multiple postures, as shown in the figure. The average error ranged from 0.33% to 3.38%.

These teams chose FBG optical sensors because they are susceptible, but the disadvantage is that they are expensive. These teams use chest or abdominal monitoring or chest–abdominal monitoring to improve sensitivity in respiratory monitoring. We use more FBG arrays to improve the system’s overall accuracy in improving respiratory monitoring for optical sensor sensors. This method is simple and practical and can effectively reduce bias. In addition, more infrared reflective markers worn by experimenters can also improve the rigor of the experiment. We should enhance the number of sensors and the quality of a single FBG sensor, which is missing from the above team. We can start with the packaging material of a single FBG sensor and improve the sensitivity of the FBG sensor through the polymer, which is an essential means to improve the sensitivity. In the breathing monitoring process, the wearer’s posture and movement status is also taken into account, which is very in line with the daily wearing requirements because people breathe differently in different postures, such as resting, sitting, lying, running, etc. These factors can be considered to improve the accuracy of wearable FBG sensors in daily monitoring. 

Different from the traditional respiratory monitoring based on chest and abdomen movement, Arjita Das’s team designed a respiratory monitoring mask based on the principle of respiratory airflow, combining FBG with an N95 mask [95] (Figure 5), which is very common in daily life, and designed a low-cost, straightforward to wear respiratory monitoring mask in daily life. Fix the silicone diaphragm with a diameter of 24 mm and a thickness of 0.8 mm on the rear cover of the valve in the plastic breathing valve of the N95 respirator. The diaphragm can be used for air filtration. Use epoxy resin to bond the FBG sensor flush to the protruding outer surface of the valve silicone diaphragm on the breathing mask. The team has a new understanding of FBG sensor breathing monitoring. First, from the monitoring site, previous FBG breathing monitoring devices start with chest and abdominal monitoring. Instead of using mouth-breathing airflow to design breathing monitoring masks, the team uses polymer packaging for FBG sensors, using epoxy and silicone diaphragms, which not only improves the wear resistance of the sensors. It also improves sensitivity. In the present trials, for a normal breathing cycle, the respiratory rate has been obtained to be 15 bpm; for a slow breathing trial, it has been obtained as 6 bpm.

### 2.2. FBG Joint Angle Monitoring

The wearable joint angle monitoring device is the most popular application of physical rehabilitation therapy. The FBG-based joint angle monitoring device can be applied to all joints of the bank y at present [96,97,98,99,100]. The continuous monitoring of the joints for a long time is an important reason for the widespread application of physical rehabilitation therapy. In traditional medical diagnosis, a hand-held goniometer is usually used for the measurement, which requires a long time to record data, and the accuracy is not high. It is time-consuming and labor-intensive, and the effect is not good. Another method uses an encoder, potentiometer, inertial magnetic unit, etc. [101]. However, they are susceptible to dislocation, the system is not compact, it requires frequent debugging, and both have the disadvantages of electrical equipment, such as electromagnetic interference [102,103]. The appearance of joint monitoring equipment based on optical fiber sensors has dramatically changed this dilemma and has received more and more attention in medical monitoring. The FBG sensor is encapsulated in a flexible material and fixed on the joint with an elastic belt. When the joint flexes and extends, the flexible material will stretch, deform, and transfer to the FBG.

That acquaintance team weaved an FBG optical fiber into the elastic fabric to continuously monitor the elbow joint [29] (Figure 6a). The equipment should have packaged the FBG. Taking into account the daily cleaning of wearable devices and the impact of perspiration on sensitivity, in the test, a subject wore the device to exercise on a treadmill for 30 min to record data, washed the device after exercise, and then recorded data after the same test. Data comparison shows that sensors maintained good linear repeatability before and after cleaning, and the Bragg wavelength displacement of the elbow angle is about 60 pm. The sensitivity is 0.5 pm/°. However, the device can only perform 0~120° flexion and extension tests, and the exposed FBG is very easy to break when performing large-scale exercise and stress flexion and extension. The knee joint angle measurement device [46] that Shweta Pant’s team proposed converts the tibiofemoral angular motion into strain changes on an organic glass cantilever beam using FBG sensors (Figure 6b). Two fiber Bragg grating sensors, made of non-hydrogenated photosensitive optical fibers, are affixed to the cantilever beam. During knee flexion and extension, any angular motion of the tibia relative to the femur causes the rotation disc to rotate correspondingly. This rotational movement of the disc applies tension to the cantilever beam through threads, resulting in strain variations that the fiber Bragg grating sensors can detect. Three sensitivity modes (high, medium, and low) can be achieved by adjusting the distance between the disc and the cantilever. In testing, the device demonstrated a maximum deflection angle of 150° in high sensitivity mode, with a sensitivity of 0.012 nm/° and a resolution of 0.083°. Martina Saltier’s team first attempted to develop a wearable FBG-based back spine monitoring system for workers [36] (Figure 6c), using 3D printing technology to convert an FBG encapsulated in a silicone material matrix to produce a flexible sensor. The highly flexible and stretchable support base improves the robustness of FBG. The wearable device consists of two elastic bands, and flexible sensors are fixed at the intersection of the bands using double-sided adhesive tape. Participants wear the device while performing back flexion and extension movements, ensuring good linear repeatability, high correlation coefficients, and sensitivity values. Subsequently, using a motion capture system as a reference, the maximum average absolute error obtained when the output displays good consistency detection is approximately 16%. The Shweta Pant team, for the first time, will combine the dynamic knee angle measurement with the measurement used to measure the front foot load dynamically [47] (Figure 6d). The new cantilever method used by the front foot load measurement equipment can ensure that the specifically applied load on the front foot area can be obtained. The knee angle measurement device is the same as the Shweta Pant team’s knee angle measurement device. Both FBG optical fibers are pasted on the rotatable plexiglass disc’s cantilever, which is fixed on the thigh and calf through the elastic belt, and the knee joint movement drives the cantilever movement to make the FBG strain on the cantilever. The sole force measuring device is a vulcanized rubber shoe with FBG on the sole. The FBG is not directly embedded in the sole but is installed on the cantilever based on glass epoxy resin so that the fixed end is located on the second sole layer on the top, and the other end is fixed on the exclusive top layer through a probe. The subjects measured the front foot load through the front foot load measuring equipment during the movement and evaluated the maximum front foot load and one/two limb supports. In particular, the swing and standing posture phases can be evaluated from knee angle measurements and forefoot load measurement equipment in the same walking cycle, which has proved consistent with this study. In addition, synchronization of load changes between the knee a forefoot can be observed.

High-precision gloves for finger joint angle are trendy in finger joint monitoring and human–computer interaction. Chandan Kumar’s JHA team designed a high-precision instrument glove to measure finger binding [70] (Figure 7a). The sensing unit comprises an FBG sensitive to the axial strain caused by finger bending. The FBG is connected to a spring (10 mm long, 0.2 mm inner diameter, and 1.9 mm outer diameter), and then the spring is bonded to the polyvinyl chloride (PVC) pipe with cyanoacrylate glue. Bending the finger will produce strain in the FBG. The sensor provides a very high angular resolution of 0.1° with a high sensitivity of 18.45 pm/°. The accuracy of 0.13° and 0.67° using mechanical setup and manual evaluation is much better than many other reported sensors. The sensor showed excellent repeatability with a maximum standard deviation of 0.30° and 0.79° on the mechanism and hand, respectively. The results are verified using a pure calibrated IMU (inertial measurement unit) sensor. Compared with IMU, the sensor also shows a better dynamic response. The maximum speed is 80°/s. However, the number of FBG sensors is insufficient. If more FBG sensors are integrated, the accuracy of the equipment can be improved. The team optimized the high-precision gloves in 2020.

FBG gloves comprise ten FBG sensor units connected to the commercial Lycra spandex gloves [48] (Figure 7b). The FBG sensor unit is constructed as follows. The optical fibers are secured inside the tube using cyanoacrylate adhesive. The spring fiber assembly is placed inside a 3D-printed tube with an inner diameter of 4 mm and a length of 34 cm. Five PVC foam strips are connected to the glove, and the sensor unit is properly fixed. The glove is sewn with a protective layer made of Lycra fabric. This minimizes the risk of damaging the FBG sensor unit during repeated glove wearing and removal. Building upon the previous design, a separator is implemented, allowing only the replacement of damaged FBG sensors. The device achieves a remarkably high angular resolution of 0.1° and compares the accuracy and repeatability of all ten glove sensors with the pure calibrated IMU sensors. This glove is superior to many sensor gloves reported earlier, with an average error of 0.80°. The standard deviation (1.01°) and range (2.60°) and the human–computer interaction between the sensing glove and the virtual VR platform have expanded the application of high-precision gloves. Jun Sick Kim’s team will use five FBG sensors to form a wearable glove module [42] (Figure 7c). FBG strain sensors will be encapsulated in a rectangular epoxy resin matrix at an offset distance of about 53 µm from the neutral axis. The sensors are rectangular and 2 mm wide. Five FBG encapsulated sensors correspond to five fingers, respectively. Due to different finger lengths, nine nodes of the FBG sensors are used for the index finger, middle finger, and ring finger, respectively; the little finger has seven nodes; the thumb has five nodes. Unlike traditional wearable glove devices, FBG strain sensors move continuously according to the hand size and the joint’s curvature to achieve high-precision real-time calculation. Typically, four algorithms are proposed to measure the inclination angle, proximal interphalangeal (PIP) joint, and metacarpophalangeal (MCP) joint angle. The maximum and minimum angle errors of the index finger are 3.53° and 1.89°, respectively, which are applicable to dip joints, respectively; 3.15°and 1.85° for PIP joints, respectively; and 2.53° and 2.11° for MCP joints, respectively. For the thumb, the maximum and minimum angular errors are 2.56° and 2.02°, respectively, for the IP connector and 3.43° and 1.41° for the MCP connector. The maximum angular error of the MCP joint of the thumb is 1.01° and 2.42°, and the angle error of the other joints is small. The average error angle of the wearable, handheld module is 0.47° ± 2.51°, and the average absolute error (MAE) is 1.63° ± 1.97°. The results show that the angles of all finger joints can be measured with high accuracy, even for different hand sizes, and the measurement accuracy varies with the algorithm.

### 2.3. FBG Other Applications

FBG-based wearable optical fiber sensors can also be used in heart rate, blood pressure, and temperature monitoring. Daniela Lo Presti’s team has developed a multi-sensor configuration based on the FBG array and encapsulated it into the skin-like soft matrix [103]. Four FBGs are encapsulated in a silicone rubber flexible matrix. Flexible sensing elements are closely attached to the chest. The sensing mechanism is based on the chest wall amplitude movement and the strain transmission between the silicone matrix through mechanical coupling. The cardiac vibration is measured from multiple locations simultaneously. The data are processed by Matlab and compared with an electrocardiogram (ECG) to obtain high consistency. Shouhei Koyama team has developed a wearable FBG blood pressure monitoring device [34]. An FBG sensor is installed at the pulsation point of the human body to measure the human pulse signal. The calibration curve is established by using the pulse wave signal and the reference blood pressure measurement value, and the blood pressure is calculated through multivariate analysis. Considering the influence of measurement height on blood pressure measurement, the team conducted blood pressure monitoring in three postures, standing, sitting, and lying, and took the standard blood pressure monitors in the market as reference instruments. Finally, the measurement accuracy of the wearable FBG blood pressure monitoring device is better at the same height. However, the measurement accuracy of the wearable FBG blood pressure monitoring device became worse with the increase in size. Hongqiang Li’s team has developed a wearable body temperature monitoring device based on FBG [5]. The FBG sensor used for temperature monitoring needs stable temperature characteristics. Therefore, exposed FBG optical fiber is not suitable for temperature monitoring. It is necessary to put the FBG into a polymer with good temperature stability.

The polymer the team uses is the copolymerization of an unsaturated polyester resin mixture containing 5.0 wt% methyl ethyl ketone peroxide (MEKP) and 2.0 wt% cobalt naphthenate at 25 °C. Unsaturated polyester resin is formed by condensation polymerization of saturated and unsaturated dicarboxylic acids and idols. This resin forms a highly durable structure and coating. The temperature sensitivity of FBG encapsulated with polymer is 150 pm/°C, almost 15 times that of bare FBG. Weave the packaged FBG sensor into the T-bar. When the body temperature is constant, the external ambient temperature will increase by 1 °C, and the FBG measurement temperature will increase by 0.04 °C. Then, the relationship between the change in external environment temperature and the evolution of FBG measurement temperature is obtained. Through testing and comparison, it is concluded that the sensitivity coefficient of the FBG sensor is 0.15 nm/°C, and the temperature error of FBG measurement is 0.18 °C.

### 2.4. Summary of FBG Wearable Medical Monitoring Equipment

Table 1 summarizes the applications of wearable FBG sensors in the field of medical monitoring, categorized in the following order: heart rate monitoring, respiratory monitoring, joint angle monitoring, and blood pressure monitoring. This section will summarize the applications of FBG sensor-based wearable medical monitoring in recent years, as shown in the table. In recent years, FBG sensor-based wearable respiratory monitoring equipment has been the most popular field, with innovative optimization of respiratory monitoring equipment every year. FBG-based wearable sensors offer unique advantages and potential applications in respiratory monitoring and joint angle measurement. Currently, wearable respiratory monitoring devices based on FBG technology primarily involve placing sensors on the chest or abdomen to measure chest wall movements or abdominal wall variations for obtaining respiratory signals. This monitoring approach can provide information such as respiratory rate, depth, and patterns, which help assess patients’ respiratory function and condition. The minimum error of respiratory monitoring for multiple postures is 0.33%. Multiple gratings can be integrated within a single optical fiber, enabling simultaneous measurement of multiple environmental parameters and enhancing the sensitivity of respiratory monitoring devices. Enhancing sensor performance can also be achieved by considering the packaging materials, such as encapsulating FBG sensors in polymers with different characteristics. While improving sensitivity, it is essential to consider the influence of external factors on respiratory monitoring devices, such as temperature, humidity, and the wearer’s body movements and positions. A novel approach to respiratory monitoring involves FBG sensing of respiratory airflow, which differs from the traditional chest or abdomen-based monitoring methods. For a normal breathing cycle, the respiratory rate has been obtained to be 15 bpm; for a slow breathing trial, it has been accepted as 6 bpm. In joint angle measurement, FBG-based wearable sensors can monitor joints such as the neck, elbow, knee, and fingers. For knee joint monitoring, the highest precision sensitivity is 0.012 nm/° and the resolution is 0.083°. For knuckle monitoring, the highest precision angular resolution is 0.1°. By monitoring the range of motion and changes in joint angles, the functional status of joints can be assessed, aiding in the diagnosis and treatment of joint disorders. Combining knee joint monitoring with gait analysis enables analysis of body posture and movements, providing guidance for motion training and rehabilitation programs. Additionally, monitoring finger joints can facilitate human–machine interaction. Due to the repetitive stretching involved in joint angle measurement, the bare optical fiber is prone to breakage. Therefore, FBG sensors need to exhibit good flexibility. Encapsulating FBG sensors in materials like silicone resin can enhance their flexibility, and integrating more gratings can significantly improve sensitivity. FBG-based wearable fiber optic sensors also face several challenges and limitations. The manufacturing and integration of FBG sensors are relatively complex, requiring precise fiber handling and grating fabrication techniques. This increases the manufacturing costs and process requirements of the sensors. The data generated by FBG fiber optic sensors is typically in the form of optical signal variations, which require data processing and interpretation to extract useful information. This necessitates advanced algorithms and signal processing techniques to handle the optical signals and convert them into practical biometric parameters such as heart rate, blood pressure, blood oxygen concentration, and others. In blood oxygen monitoring, the variation in reflected light intensity detected by a photodiode from an emitting diode is used to estimate the SpO_2_ ratio in pulse oximetry. This approach has achieved a low-cost and accurate solution, with a reduced root mean square error of 2.27 ± 0.76% and an increased Pearson correlation coefficient of 0.91 [104]. Therefore, effective data processing and interpretation pose significant challenges in the field.

## 3. Wearable Optical Fiber Sensor Based on the Principle of Light Intensity Change

Wearable optical fiber medical monitoring devices based on light intensity change are also top-rated. There are three kinds of wearable optical fiber medical monitoring devices based on the principle of fair intensity change (Figure 8). One is the intensity modulation sensor using dual optical fibers. The two optical fibers are close, connecting the light source to one of the optical fibers and the other to the photodiode [104]. The light intensity can be changed by changing the distance between the two optical fibers. A photodiode monitors the change in light intensity. This method is not standard at present. In recent years, the most popular one is the macro bending principle [105,106,107]. When the optical fiber is bent, the optical fiber will be lost, resulting in visual power loss. The bending radius will cause the light transmitted in the optical fiber core to leak into the cladding. Another method is to add a reflector in front of an optical fiber. The light intensity can also be changed by changing the distance between the optical fiber and the glass. The strain of the external environment can be reflected in the change in optical power [108,109,110,111]. Wearable optical fiber sensor based on the light intensity change principle is mainly used in the medical field in joint angle monitoring, respiratory monitoring, gait assistance, and heartbeat monitoring [112,113].

### 3.1. Joint Angle Monitoring

Based on the principle of macro bending, Eric Fujiwara’s team designed a low-cost and flexible optical fiber micro-bending sensor, which can be combined with fabric gloves to monitor the flexion and abduction of the index finger and thumb [114] (Figure 9a). Place the optical fiber in the gap of the flexible silica gel structure bending the rod, and connect the silica gel structure’s edge to the fabric glove’s surface with adhesive so that the sensor’s center is above the monitored joint. When bending the finger, it drives the silicon rod to stretch, causing the optical fiber to be squeezed and causing light attenuation of micro-bending. Therefore, the change in light intensity can be related to the joint bending angle. The device’s sensitivity is tested to be 1.80°, and the average errors of the interphalangeal and metacarpophalangeal joints are less than 5° and 7 °, respectively. However, the device’s defect is that it only monitors the bending movement of the index finger and thumb, and the monitoring bending amplitude is small. The maximum measurement angle range is not given. Rezende’s team designed a wearable portable angle measurement system based on a POF curvature sensor [64] (Figure 9b). The POF curvature sensor comprises light-emitting diodes, photodiodes, and polyethylene-coated optical fibers. The light-emitting diodes and photodiodes are connected to the two ends of the optical fiber. The POF curvature sensor is fixed to the elbow joint with an elastic band. The average difference is 5.31°, and the standard deviation is 3.71° after the subjects’ flexion and extension test. Yannick D’Mello’s team also will make a wearable elbow angle monitoring device using light-emitting diodes, phototransistors, and optical fibers [115] (Figure 9c). The difference is that the team modified the optical fibers used to significantly improve the performance of the sensor, which are made of a combination of polydimethylsiloxane (PDMS)-based silicone rubber (Sylgard 184) and PDMS-based Silicone Dielectric Gel Sylgard 527 at a volume ratio of 3:2. The high tensile strength and elasticity of the optical fiber are greatly improved. The standard deviation errors measured by the bending and stretching test of the sensor fixed on the elbow with tape are 3.525°and 4.672°, respectively. This design uses a combination of elastomer and gel to make optical fibers, significantly improving the sensor’s performance compared with ordinary optical fibers. Based on the optical fiber and reflector principle, the Min Li team designed a joint monitoring device that can be used for the elbow joint and wrist [116] (Figure 9d). The Fs-n11mn is utilized as both the light source and detector. The Fs-n11mn light source, employed for wrist joint monitoring, consists of a red LED with a wavelength of 630 nm and a reflective surface made of retro-reflective tape. The reflective surface is connected to the stepper motor via a slender rod instead of a cable, ensuring a precise correlation between fiber intensity and displacement. The transmission and reception of light employ a pair of polymer fibers, specifically lxh0501-10 (E). The design principles for elbow joint monitoring align with those for wrist joint monitoring. The sensitivity and resolution for wrist motion monitoring are 0.0296 V/° and 0.338°, respectively. As for elbow joint motion monitoring, the respective values are 0.0028 V/° and 3.6°. Jing Li’s team also will design a wearable wrist monitoring device [33] (Figure 9e). The team developed a pair of new wear-resistant and sensitivity-enhanced plastic optical fiber (POF) strain sensors, and the sensitivity of the pair of wrist monitoring sensors is higher than that of previous wrist monitoring sensors. Commercial POF (eska-sk40) is adopted. One method is to etch the periodic grating made of five grooves with a grating length of 30 mm into the POF with an engraving machine. The other method uses a fiber cutting tool to manufacture a wide polished fiber with a D-shaped polishing area with a depth of about 300 µm and a length of about 30 mm. The obtained two POF sensors are respectively embedded in the polyamide wrist guard; when the subjects wear the wristband for bending movement, the light leakage of the bending part of the bending optical fiber increases, resulting in the bending loss of optical power. After testing, it is found that the sensitivity is about 0.0176V/° and 0.0167V/°, and the bending deformation is −40 ° to +40°, far exceeding the common POF in the market.

### 3.2. Respiratory Monitoring

Retno wizardry perambulating team designed a respiratory monitoring device based on the principle of optical fiber bending loss [110] (Figure 10a). The device comprises a worn respiratory sensor and a circuit that processes optical fiber output intensity modulation to a computer. The respiratory sensor uses the standard 1.55 μm laser diode (3C link transmitter) is used as the light source, photodiode nte3303 is used as the detector, SM-SC/loc fiber is selected, and the laser and photodiode are linked at both ends of the thread and embedded into the elastic belt to form a respiratory sensor. The breathing movement of the chest and abdomen will bend and deform the optical fiber, and the deformation of the optical fiber can be monitored through the photodiode. Although the accuracy of the equipment is not high, the cost is meager, and the production is simple. Hariyanti’s team will prepare a respiratory monitoring device based on the macro bending principle [12] (Figure 10b). The respiratory sensor comprises a light-emitting diode, a photodiode (t-11-155-r), and an optical fiber. The respiratory sensor is integrated into the elastic material. During the wearer’s inhalation, the diameter of the optical fiber bending will increase, increasing the intensity of the light emitted from the optical fiber. During exhalation, the bending diameter in the fiber configuration will return to its original shape, causing the light intensity to decrease again. The light intensity change received by the photodiode is transmitted to the electronic circuit, and the optical fiber output is processed into an electrical signal. The system can measure the respiratory rate of 10–100 breaths per minute with an error of 0.25%. Yu–Lin Wan ng team also will design a respiratory monitoring device based on the macro-bending principle. However, unlike previous designs, the optical fiber sensor used by the team is a D-shaped POF sensor, which significantly improves the device’s sensitivity [24] (Figure 10c). The D-shaped POF sensor is made by removing some sides of the cylindrical optical fiber, which forms a sensitive area, to improve the sensitivity of the sensor and the linearity of signal attenuation when the optical fiber is bent. The cladding and core materials have been partially stripped in D-type POF, so Young’s modulus is low. This makes the POF sensor more sensitive. Respiratory monitoring equipment is composed of red LED if-e96e, a D-shaped POF sensor fixed on an elastic band, a photodiode if-d91 (Tempe, AZ, USA), and a microcontroller (C8051F020). When the sensor is bent due to fluctuating abdominal motion, it will cause optical power loss. The system can not only monitor the respiratory rate of the human body under different motion states but also monitor the steady-state and unsteady-state respiratory signals generated due to different physiological states, and it can eliminate noise interference. The most important thing is that the cost of the system does not exceed USD 10.

### 3.3. Gait-Assist Insole

Arnaldo g. Leal Junior’s team designed a gait-assisted insole [117] (Figure 11a). The optical fibers used in this study have a fluoropolymer coating with a thickness of 10 µm and a polyethylene outer layer for mechanical protection. The power variations in plastic optical fibers (POF) are measured using the photodiode IF-D91 (Industrial Fiber Optics). Four POF sensors are distributed within the gait-assist insole, each serving as a measurement point. During walking, various sensitive regions of the gait-assist insole experience bending, leading to changes in the refractive index of the optical fibers due to the photosites effect. Both of these factors contribute to changes in the optical power—Arnaldo G. Leal’s team also designed a gait-assist insole utilizing commercial polymethyl methacrylate (PMMA) POFs [60] (Figure 11b). The fluoropolymer coating of the POF sensors has a thickness of 10 µm and an additional polyethylene outer layer for mechanical protection. The insole base is made of TPU material using 3D printing technology, while the top layer is constructed from PLA material. The POF sensors are encapsulated within the intermediate layer of these two materials. The application of external pressure results in optical power loss. Participants wore the gait-assist insole to assess its ability to detect gait events and estimate the user’s weight, with an error of less than 3.4%.

### 3.4. Summary of Wearable Optical Fiber Sensors Based on Light Intensity Change in the Medical Field

Table 2 summarizes the applications of wearable fiber optic sensors based on optical intensity variation in the medical field, categorized in the following order: joint angle monitoring, respiratory monitoring, and gait monitoring. At present, wearable optical fiber sensors based on the change of light intensity are used in the medical field in respiratory monitoring, joint angle monitoring, and gait-assisted insoles, among which the most popular application is based on the macr-bending principle of optical fiber. Compared with the FBG-based wearable optical fiber sensor, the wearable optical fiber sensor based on the change of light intensity has the advantages of simple structure, easy fabrication, and low cost. Sensors based on light intensity changes typically consist of an optical source and an optical receiver without the need for complex optical components or fiber layouts. The optical head can be a simple light-emitting device such as an LED (light-emitting diode) or a laser diode. In contrast, the optical receiver can be a standard light detection device such as a photosensitive resistor, a photodiode, or a light sensor. This simple composition enables a relatively easy manufacturing process for the sensor without requiring complex process steps or expensive materials. The sensor achieves the transmission and measurement of light signals by connecting the optical fiber to the optical source and the optical receiver. Due to their simple structure and use of standard components, optical fiber sensors based on light intensity changes generally have lower manufacturing costs. Compared to other fiber sensors, they do not require complex optical devices or precision manufacturing processes, allowing for manufacturing and large-scale production under more economical conditions. In respiratory monitoring, wearable sensors based on optical intensity variation typically utilize fiber optic light sources and receivers, making them relatively simple and easy to manufacture and use. The minimum error is 0.25%. They often have lower manufacturing costs, making them an affordable choice. Real-time respiratory signals can be obtained by continuously measuring changes in light intensity, enabling prompt feedback and analysis. In the domain of joint angle monitoring, sensors based on optical intensity variation offer flexibility to adapt to different joint monitoring requirements. This can be achieved by adjusting the positions and angles of the light source and receiver to accommodate various joint configurations. For knuckle monitoring, the maximum sensitivity and resolution were 1.80°, 0.0296 V/° and 0.338°, respectively. For elbow motion monitoring, the respective values are 0.0028 V/° and 3.6°. However, sensors based on optical intensity variation may be susceptible to interference from external light sources and ambient light, and they are more sensitive to environmental noise and temperature changes. It is crucial to select stable, uniform, and appropriately spectrally ranged light sources to improve the performance of optical sensors based on intensity variation. The quality and stability of the light source directly affect the sensor’s performance. Additionally, temperature compensation techniques can be employed to reduce the impact of temperature variations on the measurement results. Proper encapsulation and protection measures are essential to enhance the durability and stability of the sensor. Choosing suitable encapsulation materials and techniques ensures the reliable operation of the sensor under various environmental conditions.

## 4. Wearable Optical Fiber Medical Monitoring Equipment Based on FPI

The FPI sensor, based on the principle of Fresnel reflection, utilizes the optical interference phenomenon between a pair of fiber optic end faces to enable measurements. An air cavity is formed between the fiber optic end faces, where the incident light enters the hole and undergoes multiple reflections inside, resulting in numerous reflected beams. These reflected rays then pass through the second end face of the hole and eventually refract back into the fiber. The coherent interference between these reflected and transmitted light beams generates an interference signal. The measurement of physical quantities with the FPI sensor relies on the variation of interference light intensity. By monitoring the changes in interference light intensity, one can deduce the variations in the physical quantities. The intensity distribution (I) of the FPI sensor is determined by the length of the microcavity (L) and the refractive index (n) of the cavity material. Equations (1) and (2) describe this dependency:(4)$$I=I_{1}+I_{2}+\cos(\delta )\sqrt{I_{1}I_{2}}$$
(5)$$\delta =\frac{4\pi nL}{\lambda }$$

I1 and I2 are the intensities of the reflected optical signal in the two mirrored surfaces of the FPI, and δ represents the round trip optical phase difference between two adjacent wavelength signals, λ, at a normal angle of incidence. Therefore, any change in the micro-cavity length/volume or refractive index induces a shift in the reflected interference.

Currently, only one wearable optical fiber medical monitoring device based on FPI needs to be expanded by R&D personnel. Although FBG is an accurate, reliable, and widely explored sensing technology, it may increase the production cost of sensors and the corresponding high-cost interrogation system. Using cost-effective FPI fiber optic sensing solutions is a reliable alternative to these applications.

Maria de F Á TIMA domains team will develop an ankle joint monitoring device based on FPI for the first time [23] (Figure 12). The damaged optical fiber showed a series of periodically spaced gaps in its core, which prevented the transmission of optical signals. When the optical fiber is spliced into the standard single-mode optical fiber (SMF), the initial gap will form a more significant gap. The optical fiber is cut and joined to the SMF fiber to make a single in-line FPI microcavity. The applied strain causes the thread to elongate, which leads to an increase in microcavity length and wavelength shift. The quartz optical fiber encapsulated with epoxy resin is integrated into the elastic band with acrylate adhesive, and the elastic band is fixed on the ankle joint. With the flexion and extension of the ankle joint, the elastic band will be stretched, and proportional tension will be induced in the epoxy resin structure. FPI modulates the spectral wavelength shift of the optical signal. After testing, the sensitivity of the equipment is 0.0296 ± 0.001 mw/°.

## 5. Conclusions

This article reviews the applications of wearable optical sensors in the medical field. In three chapters, wearable optical sensors based on FBG, wearable optical sensors based on light intensity changes, and wearable optical sensors based on FPI are introduced. Respiratory and joint angle monitoring are widespread applications of wearable optical sensors in the medical field. The FBG-based wearable optical sensor is currently the most widely used one. It has high accuracy in respiratory and joint angle monitoring, but its cost is high. Wearable sensors based on changes in light intensity are low in charge and easy to fabricate but with low accuracy. Compared with the first two, wearable optical fiber sensor based on FPI has few applications. However, it has high research and development potential because it is less expensive than FBG, more accurate, and has significant economic benefits. Wearable optical sensors have great potential in the future medical field. Presently, wearable optical sensors have a relatively single application in the medical field, which can expand to the direction of heart rate, blood pressure, pulse, etc. Developing susceptible polymer fibers can be considered when improving the accuracy of wearable optical fiber sensors. By choosing different light cores, different coatings of other materials, and using 3D printing technology in the packaging process, the fabrication process can be made more accessible. These factors will significantly enhance the optical properties of the sensor. While improving the sensor’s accuracy, we should also consider the compact, compact, and comfortable size to make it more suitable for the needs of users in daily life and to adapt to various environments, such as being wet, sweating during running, etc.

## 6. Recommendation and Challenge

In the Internet of things era, the combination of wearable optical sensors and wireless technology will be a prevalent application for remote monitoring in the future. Data transmission to professional medical service stations, supervised by professional doctors, will be a prevalent application of remote medical monitoring in the future. The number of patients going to hospitals and medical stations will be significantly reduced, as will the costs. Wearable optical sensors have great potential in the medical field and will be an essential way to change the medical environment in the future. Wearable optical fiber sensors in medical monitoring face various challenges that must be addressed. One key challenge is ensuring comfortable and accurate attachment to the body of patients or users, enabling precise measurements. To enhance comfort and wearability, it is recommended to manufacture sensors using soft, breathable, and skin-friendly materials and user-friendly wearing methods such as adjustable straps or patches. Another challenge arises from potential interference sources, including environmental factors, which can introduce errors in sensor measurements. Implementing effective signal processing techniques becomes crucial to mitigate these issues. This involves employing filtering and noise reduction algorithms to minimize interference’s impact and enhance the measured signals’ reliability. Furthermore, it is essential to focus on designing circuit layouts and protective measures to minimize external signal interference. By carefully considering the form and implementing appropriate shielding techniques, the sensors can be safeguarded against external disturbances, ensuring more accurate and reliable measurements. Ensuring sensor data’s accuracy and consistency is paramount in medical monitoring. Thorough calibration and validation processes should be undertaken to establish a strong correlation between the sensor measurements and the results obtained from standard instruments or established methods. Regular calibration and maintenance of the sensors are necessary, along with recording performance indicators and accuracy metrics, to ensure ongoing data accuracy and reliability in medical applications.

## Figures and Tables

**Figure 1 sensors-23-06671-f001:**
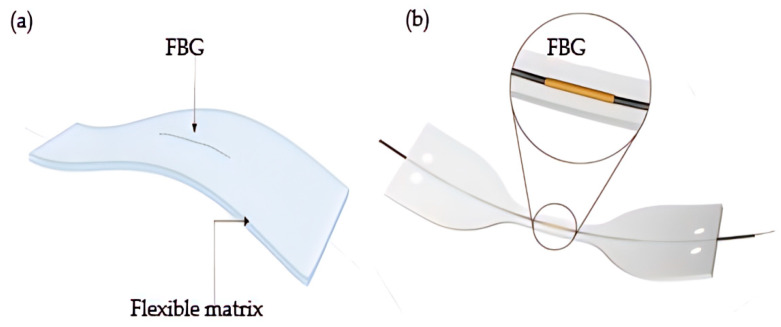
FBG encapsulated in a flexible material (**a**) and FBG encapsulated in dumbbell silicone (**b**).

**Figure 2 sensors-23-06671-f002:**
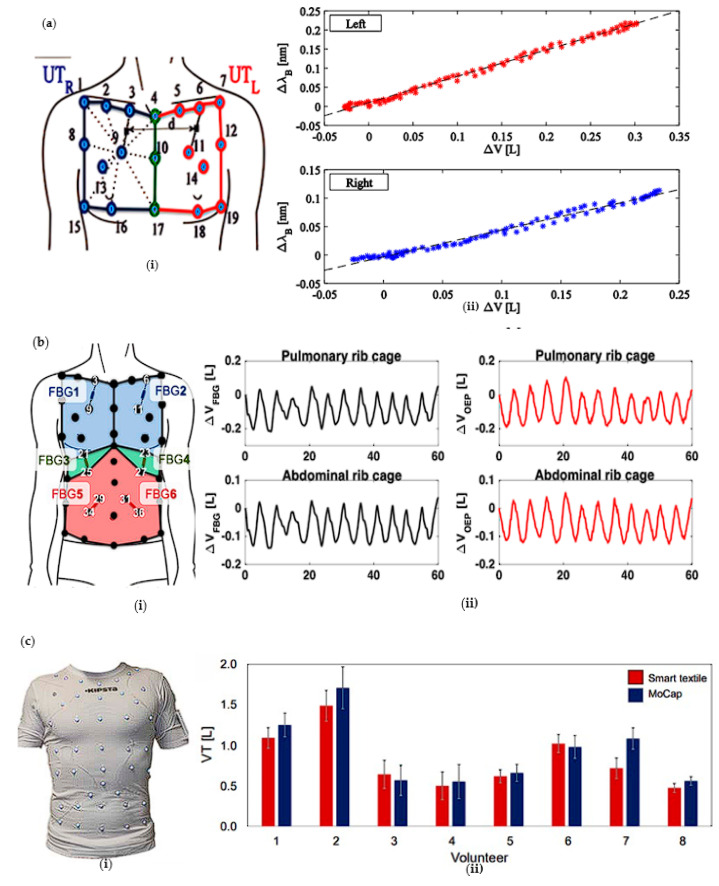
(**a**) Schematic diagram of chest breathing monitoring (**i**) and correlation between FBG wavelength change and UT volume (L) (**ii**) [63]; (**b**) schematic diagram of thoracoabdominal breathing monitoring (**i**) and FBG volume (L) change (**ii**) [50]; (**c**) the schematic diagram of chest and abdomen respiratory monitoring (**i**) and the VT value calculated by smart textile are compared with the VT value collected by motion (**ii**) [51].

**Figure 3 sensors-23-06671-f003:**
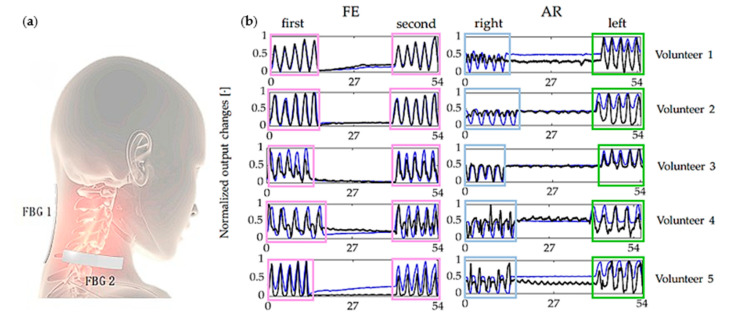
The shoulder neck monitoring structure diagram (**a**) and the output changes of wearable (black line) and motion capture (mobcap) systems (blue line) collected during shoulder neck up, down, left, and proper repetition (**b**). The flowmeter (blue line) and FBG (black line) are the signals collected by each volunteer during quiet breathing (**c**) and rapid breathing (**d**) [45].

**Figure 4 sensors-23-06671-f004:**
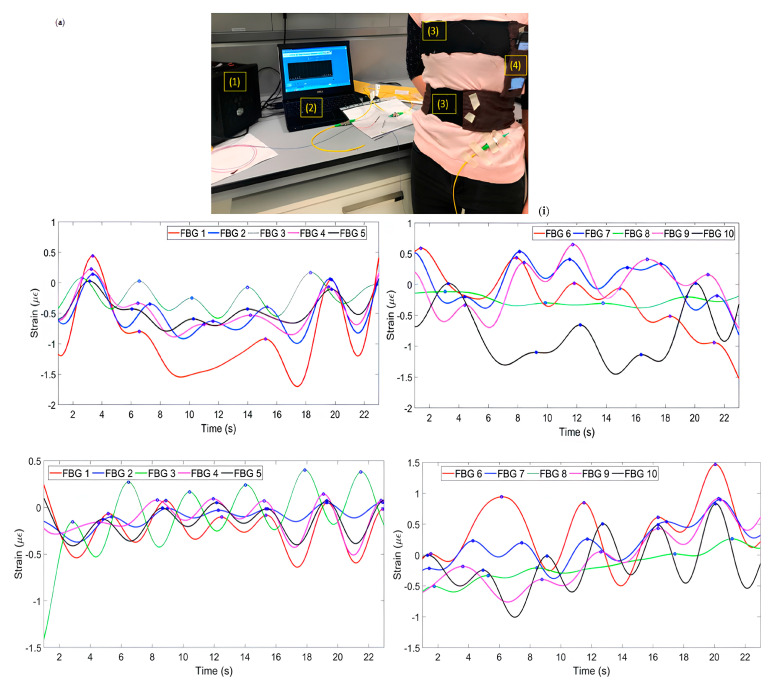
(**a**) Schematic diagram of thoracoabdominal breathing monitoring and 5 FBGs of the chest (**i**) and abdomen (**ii**) detected strain patterns under four states of standing, sitting, lying down, and running [22]; (**b**) schematic diagram of thoracoabdominal breathing monitoring (**i**) and strain detected by FBG under various postures [47] (**ii**).

**Figure 5 sensors-23-06671-f005:**
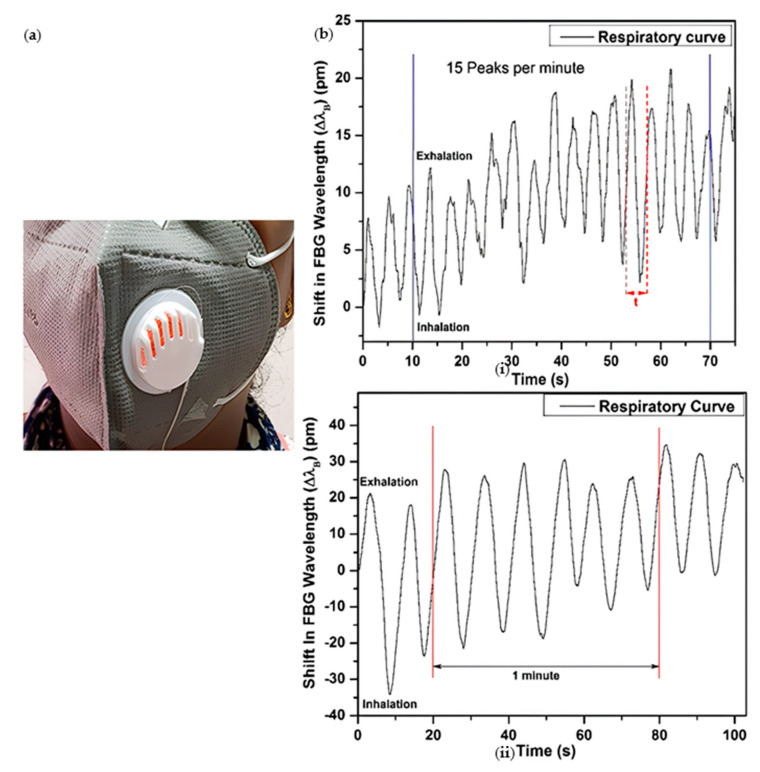
(**a**) Schematic diagram of respirator for respiratory monitoring (**b**) detection of the respiratory curve and peak wavelength drift in normal respiratory cycle (**i**) and slow respiratory cycle (**ii**) [95].

**Figure 6 sensors-23-06671-f006:**
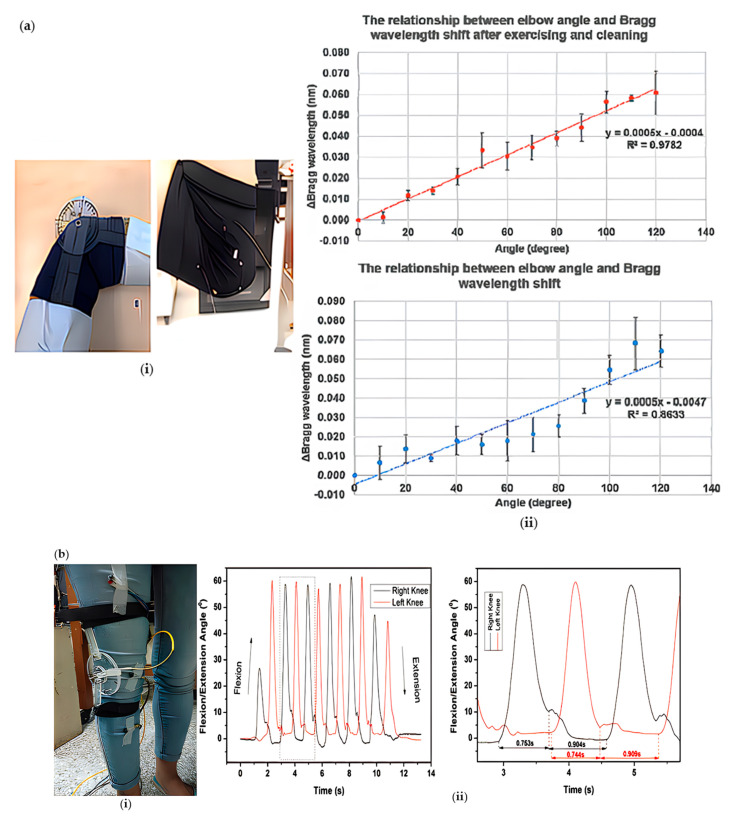
(**a**) Schematic diagram of elbow joint monitoring (**i**) and the relationship between elbow angle and Bragg wavelength displacement before and after movement (**ii**) [29]; (**b**) schematic diagram of knee joint monitoring and kind response during walking (**i**), enlarged image of a gait cycle (**ii**) [46]; (**c**) lumbar monitoring schematic diagram (**i**) and wearable output ${\;∆\lambda }_{B}$, distance between L1 and L3, lumbar angle obtained according to e θ trend (**ii**) [36]; (**d**) gait monitoring diagram (**i**) and simultaneous response of kids and fluids during walking (**ii**) [47].

**Figure 7 sensors-23-06671-f007:**
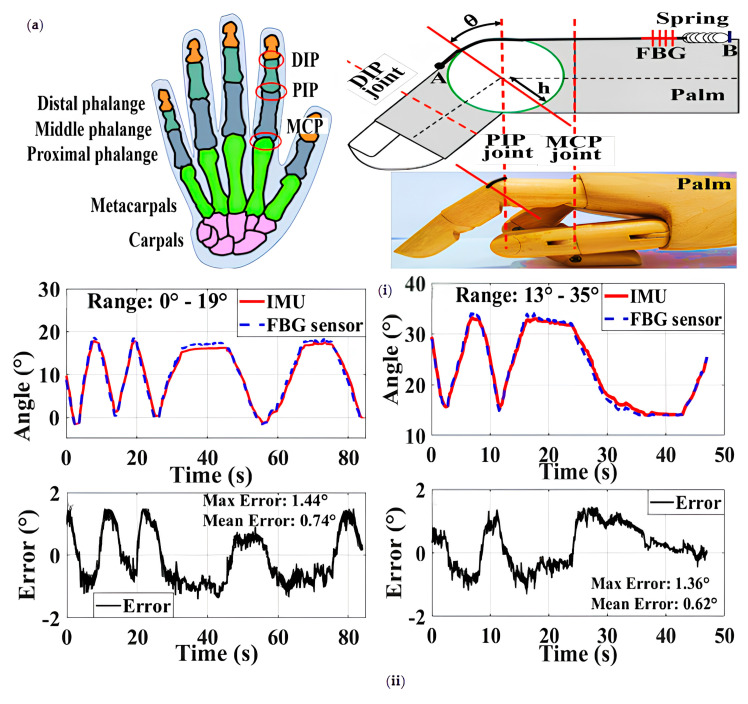
(**a**) Structural diagram and dynamic response of FBG sensor in the range of 0–19° (**i**) and 13°–35° (**ii**) compared with IMU [70]; (**b**) structure diagram (**i**) and angle results measured by MCP joint (upper) and PIP joint (lower), FBG sensor and IMU sensor (**ii**) [48]; (**c**) structure diagram (**i**) and the length of measured angle and actual angle (measured by goniometer) relative to index finger: 117 mm, 97 mm, 77 mm. Measure the linearity of the MCP joint angle according to the algorithm and finger length (**ii**) [42].

**Figure 8 sensors-23-06671-f008:**
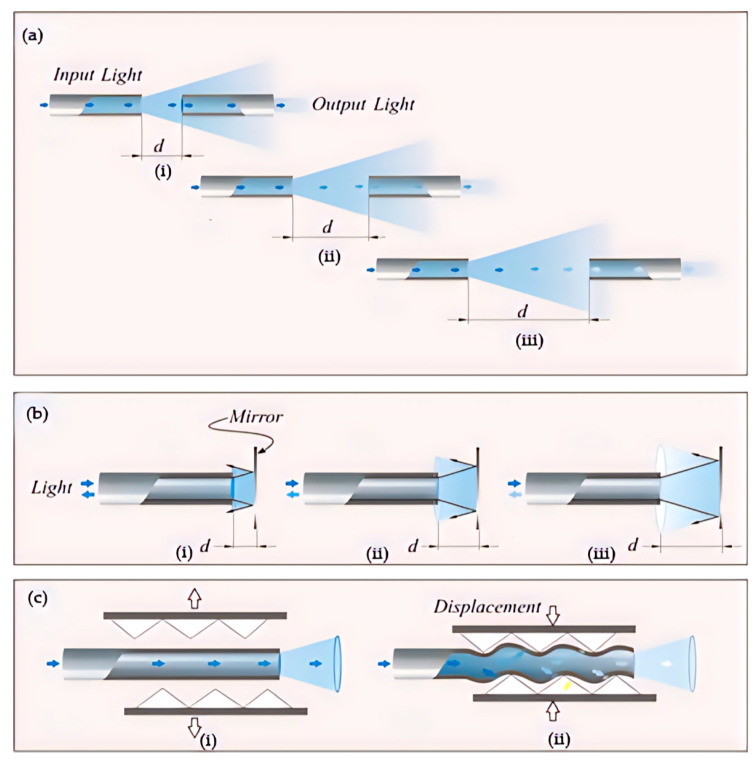
(**a**) the principle of changing light intensity is changing the distance between two optical fibers and schematic diagram of the process of light passing through an optical fiber (**i**–**iii**); (**b**) changing the light intensity by changing the direct distance between the optical fiber and the mirror and schematic diagram of the process of light passing through an optical fiber (**i**–**iii**); (**c**) changing the light intensity principle by macro bending of optical fibers and schematic diagram of the process of light passing through an optical fiber(**i**,**ii**).

**Figure 9 sensors-23-06671-f009:**
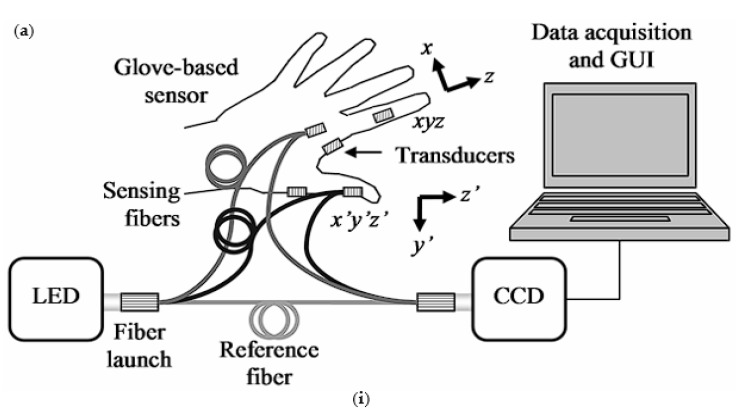
(**a**) Finger joint monitoring structure diagram (**i**) and sensor response to angular displacement change of finger joint (**ii**) [114]; (**b**) elbows joint monitoring structure diagram (**i**) and joint angle measurement results (**ii**). The blue line represents the potentiometer and the red line represents POF [65]; (**c**) wrist monitoring structure diagram (**i**) and measurement response of wrist flexion, extension, and abduction adduction motion (**ii**) [115]; (**d**) elbow joint monitoring chart (**i**) and average optical response of optical fiber during elbow bending of art (**ii**) [116]; (**e**) elbow and wrist joint monitoring chart (**i**) and output voltage increasing with carpometacarpal flexion angle (**ii**) [33].

**Figure 10 sensors-23-06671-f010:**
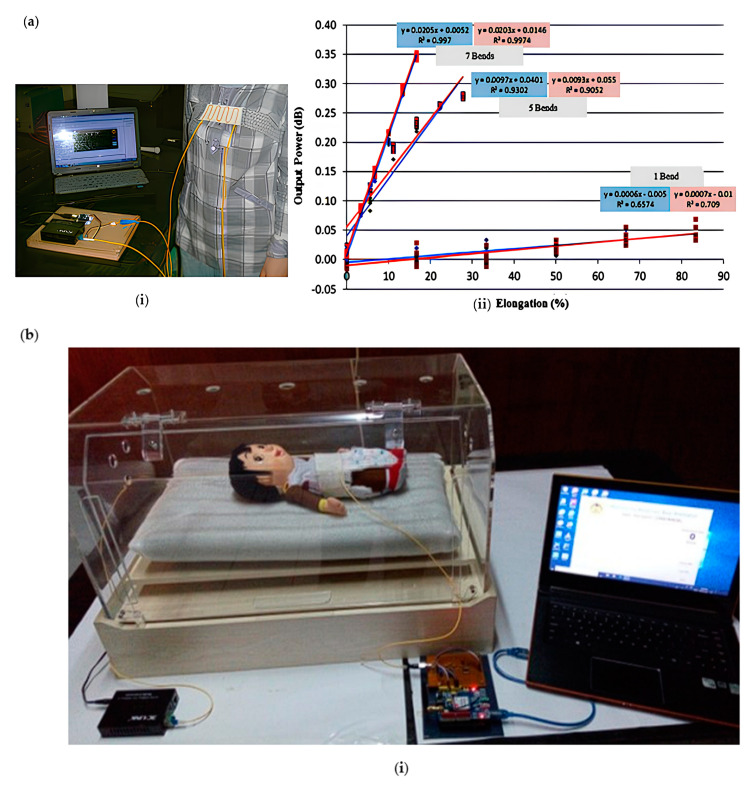
(**a**) Structure diagram of respiratory monitoring (**i**) and function of optical output power (1 cm in diameter) of respiratory sensor and elongation of respiratory sensor (**ii**) [110]; (**b**) respiratory monitoring structure diagram (**i**) and power output of optical fiber respiratory sensors with different bending times and different bending diameters (**ii**) [12]; (**c**) the time domain waveforms of respiratory monitoring structure diagram (**i**) and (the first three columns) original signals at rest, walking and running, respectively. (Middle three columns) time domain waveform of the respiratory signal after signal and processing. (Last three columns) frequency domain corresponding to respiratory signals ((**ii**) [24]).

**Figure 11 sensors-23-06671-f011:**
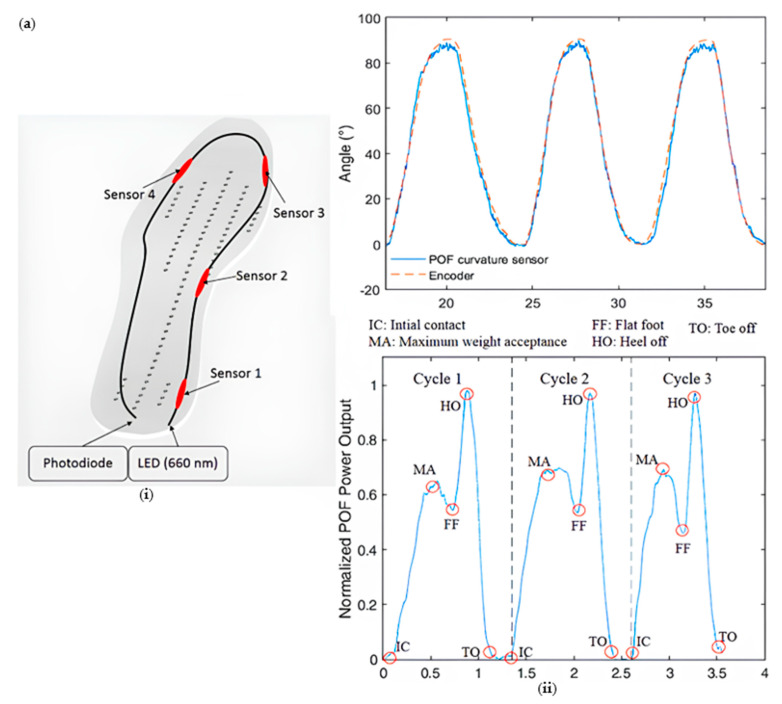
(**a**) Gait-assist structure diagram (**i**) and POF curvature response of knee flexion and extension cycle of modular exoskeleton (above), POF insole for functional electrical stimulation (FES) assisted-gait phase detection (below) (**ii**) [23]; (**b**) gait help structure diagram (**i**) and ground reaction force (GRF) and plantar pressure of female subjects weighing 46kg (above). The solid line is the average GRF, and the shadow curve is the standard deviation of 5 cycles; (below) GRF and plantar pressure measurements are standardized for all 20 participants. Solid lines represent the mean GRF and shaded curves represent the standard deviation (**ii**) [60].

**Figure 12 sensors-23-06671-f012:**
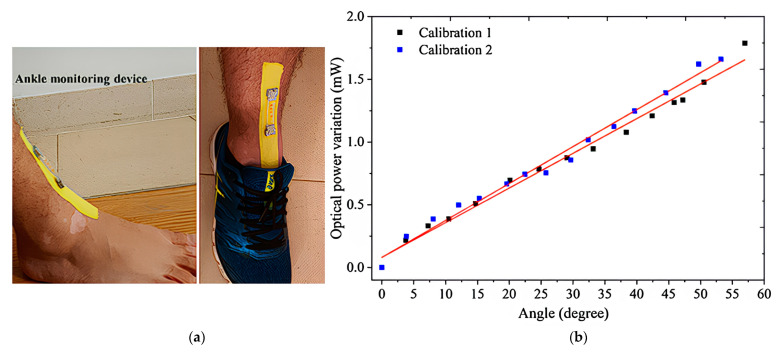
Wearable ankle monitoring device based on FPI (**a**) and optical power variation with angular displacement. Points are experimental data, and straight lines correspond to linear fitting (**b**) [23].

**Table 1 sensors-23-06671-t001:** Application of wearable FBG sensor in the medical field.

Working Mechanism	Application	Packaging Materials	Characteristics	REF
FBG	Heart rate monitor	Flexible silicone rubber matrix	Multiple FBG matrices; multipoint measurement	[37]
FBG	Breath	Silicone rubber	Sensitivity is always higher than 0.66 and 0.35	[62]
FBG	Breath	Silicone	Simple structure; cost is low	[50]
FBG	Breath	Elastic fabric	The bias of 0.01 s	[51]
FBG	Monitor neck movement and breathing	Silicone matrix	The percentage error is 1.90%	[45]
FBG	Monitor chest and abdominal breathing	Elastic belt	Respiratory pattern reconstruction Multipoint sensing	[22]
FBG	Breath	Silicone; PI coating	The average error ranged from 0.33% to 3.38%	[47]
FBG	Breathing	Epoxy resin	Low cost; easy to wear	[95]
FBG	Joint angle	Elastic fabric	The sensitivity is 0.5 pm/°	[29]
FBG	Knee joint	Bare fiber	Sensitivity of 0.012 nm/°	[46]
FBG	Lumbar spine	Silicone	First wearable lumbar spine monitoring	[36]
FBG	Knee joint	Epoxy resin	The horse cantilever	[100]
FBG	Finger joint angle	Elastic fabric	Angular resolution is 0.1°	[70]
FBG	Finger joint angle	Rectangular epoxy resin	The maximum angle error is 4.6°	[42]
FBG	Finger joint angle	PVC	The average error is 0.8°	[48]
FBG	Blood pressure	Elastic fabric	Many positions; not affected by the height	[34]
FBG	Body temperature	Unsaturated polymer resin	Increased sensitivity by 15 times	[5]

**Table 2 sensors-23-06671-t002:** Application of wearable optical fiber sensor based on light intensity change in medical field.

Working Mechanism	Application	Packagin Materials	Characteristics	Ref.
Macroscopic bending	Finger joint	Silica gel	Sensitivity 1.80 °	[114]
Macroscopic bending	Elbow joint	Polyethylene	The mean difference is 5.31° and the standard deviation is 3.71°	[64]
Macroscopic bending	Elbow joint	Elastomers and gels	The standard deviation error is 3.525°	[115]
Optical fiber and mirror	Elbow and wrist	POF	Resolution is 0.338°	[33]
Macroscopic bending	Wrist	POF	The sensitivity is about 0.0176/°	[116]
Macroscopic bending	Breathe	Bare leak	Low cost and easy fabrication	[110]
Macroscopic bending	Breathe	Bare leak	The margin of error is 0.25%	[12]
Macroscopic bending	Breathe	D-shaped POF	Low cost and simple design	[24]
Macroscopic bending	Gait-assist insole	Fluoridated polymer cladding and polyethylene cladding	POF curvature sensor has low angle error	[117]
Macroscopic bending	Gait-assist insole	Fluoridated polymer cladding and polyethylene cladding	Error less than 3.4%	[60]

## Data Availability

Not applicable.

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
