# Peer review of "Wearable Optical Fiber Sensors in Medical Monitoring Applications: A Review"

_sensors, 2023, doi:10.3390/s23156671_

Round 1

Reviewer 1 Report

Wearable optical sensors have important applications in the medical field. This work summarizes the latest research progress of wearable optical sensors in the medical field. Three different types of sensors are introduced, including FBG type, light intensity type and FPI type. The principle and research progress of those sensors are introduced detail. The language of the article is relatively fluent and accurate. However, this article also has many problems, such as the logic of the introduction, the processing of figures and other details. The details are as follows:

1.      The logic of the introduction is a bit confusing, the paper first introduces the research background and significance of wearable optical sensors, then introduces the research background and significance of optical sensors, and finally returns to wearable optical sensors. Besides, there is duplication of descriptions in parts of the introduction.

2.      The figures in the paper are not handled well, for example, the figures are very blurry and cannot be recognized, some figures are not described accordingly in the paper, and the figures are also not clearly marked.

3.      Some of the formula symbols in the paper are confusing and the symbols that appear in the formula are fully described, and the reference order is also confused.

4.      The introduction of the research work in the recent year is very detailed, but the summary is not well.

5.      The paper only introduced three commonly used sensor types, and there are many types of sensors that are not introduced, It is not comprehensive enough.

Author Response

We acknowledge the reviewer for this comment. Please see the attachment.

Reviewer 2 Report

The authors presented a review article about wearable optical fiber sensors in medical monitoring applications. Though many previous published articles were involved, the differences of this review compared to other published reviews were not clearly addressed. Thus, this manuscript should be thoroughly improved before resubmission. The detailed comments are below:

1.   At the end of section 1, the aim and the structure of this review should be given.

2. In the introduction section, the introduction to the "Optical sensor" could be moved before the introduction to the "Optical fiber sensor" to be logical.

3. For the presentation of previous works, the words could be more concise with introduction, comparison, conclusion and discussion. 

4.  Figure 3 was not totally introduced.

5. The meaning of the abbreviations were not known, such as "UT volumes".

6. For line 414-448, not reference was inserted.

7. The principle of FPI should be added.

There are many mistakes in this manuscript.

For example, "Young people sit for a long time, work overtime, stay up late, work and study, and improper sitting posture will induce shoulder[16,17], neck[17,18], and waist disease[19-21]." Though one can understand, it needs to be corrected.

"L.P. et al." should be replaced by "the family name of the first author et al."

"Retno wizardry perambulating team will design a respiratory monitoring device based on the principle of optical fiber bending loss." “will” is not correct because it happened before.

"which needs to be expanded by R & D personnel." This sentence in line 644 is difficult to understand.

Author Response

(The authors gave the same response as above.)

Reviewer 3 Report

Principal comments:

        There is no explanation of the criteria according to which the individual items were sorted (sorted) – e.g. Table 1 (not a year or reference number).

        dtto Table 2 – “This section summarizes some medical applications of wearable optical fiber sensors based on light intensity changes in recent years.” – It is therefore according to “light intensity has the advantages of simple structure, easy fabrication, and low cost” – not clearly explained in the whole manuscript.

          Conclusion: Since fiber-optic sensors are relatively expensive, should the accuracy and achieved parameters be compared with cheap optical sensors working on the transmission/reflective principle – measuring of PPG/SpO2 signals?

          Finally, give some recommendations (instead of "Expectation").

          English seems to be OK (I’m not expert for English), but minor errors in the text (+formatting problems) – spoil otherwise a good impression of reading the article.

Some technical recommendations and asks:

        Page. 5 – here is missing any explanation about placement and used measurement strategy of the “FLOW sensor” (flowmeter) mentioned in the caption of Fig. 2.

        Fig. 2 – bad/ incorrect marking in the label it is not explained what the graphs (ii) mean specifically, what are the units "L". The text then lacks direct links to images - or some incomprehensible marking system??

        P. 11 – instead of the character "°" write deg. (to indicate the angle?)

        Fig 5. Error in caption a) is a photo, both graphs are b) - then (i) and (ii) marking to use here.

        Fig. 6 b,c – Incorrect x/y ratios for photos

        Fig. 10b, (ii) –illegible axis labels (to small fonts)

        Bibliography sources of conference papers ALWAYS skip the date to the next line.

Author Response

(The authors gave the same response as above.)

Reviewer 4 Report

Current manuscript entitled “Wearable Optical Fiber Sensors in Medical Monitoring Applications: A Review” by “Zhang et al” reviewed the latest evolution of wearable optical sensors in the medical field. Three types of wearable optical sensors are analyzed: wearable optical sensors based on FBG, wearable optical sensors based on light intensity changes, and wearable optical sensors based on FPI. The working principle of three types of wearable optical sensors is introduced, and their applications in respiratory monitoring and joint angle monitoring are summarized. Wearable optical sensors offer viable technology for prospective continuous medical surveillance and will change future medical benefits. The manuscript needs the following changes.

1.      Provide the full form of FBG and FPI, in the abstract section.

2.      Clear statements of the novelty of the work should also appear briefly in the Abstract and Conclusions sections.

3.      Improve the image quality of the figures.

4.      Discuss about the relevant works in the introduction section

5.      Introduction seems weak, authors should discuss these articles on the on-site sensors. Advances in optical-sensing strategies for the on-site detection of pesticides in agricultural foods. Polymer Optical Fiber Sensors in Healthcare Applications: A Comprehensive Review. Portable electrochemical sensing methodologies for on-site detection of pesticide residues in fruits and vegetables.

6.      In table 1 remove the Year column

7.      Provide the challenges that are currently facing with the Wearable Optical Fiber Sensors in Medical Monitoring.

 Minor editing of English language required

Author Response

(The authors gave the same response as above.)

Round 2

Reviewer 2 Report

The comments are well answered and I suppose this paper could be accepted.

English is OK.

Author Response

We acknowledge the reviewer for this comment.

Reviewer 4 Report

Accept

Author Response

(The authors gave the same response as above.)
